# Optimizing PET Glycolysis with an Oyster Shell-Derived Catalyst Using Response Surface Methodology

**DOI:** 10.3390/polym14040656

**Published:** 2022-02-09

**Authors:** Yonghwan Kim, Minjun Kim, Jeongwook Hwang, Eunmi Im, Geon Dae Moon

**Affiliations:** 1Department of Advanced Materials R&D Center, Dae-Il Corporation (DIC), Ulsan 44914, Korea; kyh871111@dicorp.co.kr (Y.K.); jwhwang@dicorp.co.kr (J.H.); 2RIKEN Center for Emergent Matter Science (CEMS), Saitama 351-0198, Japan; minjun.kim@riken.jp; 3Dongnam Division, Korea Institute of Industrial Technology (KITECH), Busan 46938, Korea; eunmi47@kitech.re.kr

**Keywords:** PET, depolymerization, catalyst, oyster shell, glycolysis, response surface methodology, box-behnken design

## Abstract

Polyethylene terephthalate (PET) waste was depolymerized into bis(2-hydroxyethyl) terephthalate (BHET) through glycolysis with the aid of oyster shell-derived catalysts. The equilibrium yield of BHET was as high as 68.6% under the reaction conditions of mass ratios (EG to PET = 5, catalyst to PET = 0.01) at 195 °C for 1 h. Although biomass-derived Ca-based catalysts were used for PET glycolysis to obtain BHET monomers, no statistical analysis was performed to optimize the reaction conditions. Thus, in this study, we applied response surface methodology (RSM) based on three-factor Box–Behnken design (BBD) to investigate the optimal conditions for glycolysis by analyzing the independent and interactive effects of the factors, respectively. Three independent factors of interest include reaction time, temperature, and mass ratio of catalyst to PET under a fixed amount of ethylene glycol (mass ratio of EG to PET = 5) due to the saturation of the yield above the mass ratio. The quadratic regression equation was calculated for predicting the yield of BHET, which was in good agreement with the experimental data (R^2^ = 0.989). The contour and response surface plots showed the interaction effect between three variables and the BHET yield with the maximum average yield of monomer (64.98%) under reaction conditions of 1 wt% of mass ratio (catalyst to PET), 195 °C, and 45 min. Both the experimental results and the analyses of the response surfaces revealed that the interaction effects of reaction temperature vs. time and temperature vs. mass ratio of the catalyst to the PET were more prominent in comparison to reaction time vs. mass ratio of the catalyst to the PET.

## 1. Introduction

Polyethylene terephthalate (PET) is a widespread and versatile thermoplastic polymer. Since PET has light weight, excellent chemical/thermal/mechanical properties, and a low price, it is widely used in packaging materials, such as films, fabrics or various types of containers. According to a recent report, global PET consumption reached 23.5 million tons in 2016 and future growth in PET packaging is expected to reach 27.1 million tons in 2025 [1]. Furthermore, it is estimated that approximately 70 million barrels of oil are used in the processing of virgin polyester fibers [2]. Unfortunately, only 9% of the plastic waste in the world has been recycled, and natural resistance to degradation has serious impacts on the environment and society by producing overwhelming waste of resources [3,4]. In order to solve the disposal of plastic waste, several methods for recycling have been developed, which are classified as energy recovery, physical recycling, and chemical recycling [5,6,7,8]. The chemical recycling of PET waste may be a substantially sustainable and distinctive method because plastic could be converted into pure value-added products by chemical processes [9,10,11].

Several chemical recycling methods have been suggested to depolymerize PET into monomers through hydrolysis, methanolysis, aminolysis, and glycolysis. Among these methods, glycolysis is the most efficient recycling method due to its low-cost process, low-volatility solvents, continuous production feasibility, and mild reaction conditions which can be processed at 180–260 °C at atmospheric pressure [7,12]. The glycolysis of PET is a form of molecular-level depolymerization by transesterification using PET ester groups and a diol with a mainly excessive amount of ethylene glycol (EG) in order to obtain the monomer bis (2-hydroxyethyl) terephthalate (BHET). During glycolysis reactions, the bonding of ester linkages for PET is broken and a monomer, BHET, is created, according to the following stoichiometry:$${\mathrm{PET}}_{n}+(n-1)\mathrm{EG}\;⇆\;n\mathrm{BHET}$$

However, PET glycolysis is an extremely slow process without a catalyst [13]. Therefore, highly active transesterification catalysts must be added into the reaction medium, such as metal salts [14,15], urea [16], solid acids [17], ionic liquid [18,19], enzymes [20], and nano-materials [21]. Metal acetates (zinc, manganese, cobalt, and lead) were mainly investigated to determine how to enhance the reaction of PET depolymerization and the yield. However, heavy metal catalysts require complicated processes to cleanse the product through separation from the reaction mixture [22]. As alternatives to these methods containing environmental and processing problems, eco-friendly catalysts such as Ba(OH)_2_, Na_2_CO_3_, and NaHCO_3_ were studied by reducing the long hours of reaction through microwave heating [23,24].

Most studies have focused on the yield of products by using the one-factor-at-a-time (OFAT) experiment method, according to which only one factor is changed while keeping others fixed [25]. However, single-factor analysis is insufficient for optimal conditions because the interaction effects between the main factors are neglected. This could lead to inadequate conclusions without considering the interactions between the key variables. To understand the effect of independent variables and their interactions for PET glycolysis, a few researchers have utilized response surface methodology (RSM), which is the mathematical and statistical method to optimize the response [26,27]. A regression model based on the key factors and the interactions between them can be used to forecast the response and choose the optimal conditions in PET glycolysis [28].

Yunita et al. have demonstrated that biomass-derived catalysts, such as calcium-based metal oxide made from food waste (eggshells and seafood shells), can also be used in the PET depolymerization process to produce BHET [29]. They discovered that the yield of BHET using a catalyst from the food waste was comparable to the commercial zinc acetate catalyst. These findings suggested that Ca-based catalysts have potential for application in the depolymerization of PET. However, statistical studies of the optimization of reaction conditions on the type of by-product were insufficient. Furthermore, it is challenging to collect and process by-products of ostrich and chicken egg shells considering the further scale-up of the resources in industrial usage. Among the by-products from biomass, oyster can be another resource due to its large production in various countries, including South Korea. The annual production of oyster in southern coastal areas is over about 280,000 tons, resulting in a large amount of oyster shell waste. Thus, oyster shell waste has caused huge social and environmental issues in this region due to the absence of a method or technique to deal with its by-products [30]. Very recently, researches have attempted to determine uses for oyster shell wastes, such as construction materials, fertilizers, stabilizers, absorbents, and as catalysts for the biodiesel and transesterification process. Nevertheless, oyster waste remains an unsolved issue. As an alternative, post-treated oyster-shell-based on calcium oxide is beneficial as a catalyst in glycolysis for PET. The continuous annual production of oyster waste enables enough supply for the processing of Ca-based catalysts in PET glycolysis reactions instead of throwing them away near residential areas, causing environmental problems.

In this study, oyster shell waste was utilized as an eco-friendly and economically-competitive catalyst in PET glycolysis. Box–Behnken design (BBD) experiments via RSM (three factors) were carried out to predict the response by constructing an adequate mathematical model. The interactive effects of three independent variables (reaction time, temperature, and mass ratio of catalyst:PET) were also investigated for optimizing the BHET yield. Thus, the complete effects between process parameters are provided for energy- and material-efficient plastic recycling.

## 2. Materials and Methods

### 2.1. Materials

PET pellets were provided by Korea Plastic Single Materials Association, Korea. The pellets were cut into 10 mm × 12 mm and cleaned by 1 M NaOH, followed by drying at 80 °C for 12 h. EG was purchased from DAEJUNG CHEMICALS & METALS CO. LTD., Sinan, Korea and used without further purification. Oyster shell waste was obtained from a commercial oyster farm in Tongyeong city, southern Korea.

### 2.2. Preparation of Catalysts

The oyster shells were washed and dried at 100 °C for 3 h in order to remove organic foreign material on the shell surface. The shells were crushed using a hammer and then mechanically milled by using a GLBM-G ball mill (GLOBAL LAB, Siheung, Korea) with zirconia balls at a rotational speed of 300 rpm for 24 h. Particles of sizes less than 100 μm were obtained by using the fine sieving method. In order to transform the phase, the shells were heated at 1000 °C in the air with a ramping rate of 20 °C min^−1^ for 5 h. Produced catalysts were stored in a vacuum to keep them from moisturizing.

### 2.3. Glycolysis Mechanism

Glycolysis is a depolymerization process, according to which PET is transesterified in an excessive EG to yield oligomer and EG. In our study, the oligomers were continuously transesterified to produce BHET according to the following chemical reaction (Figure 1):

### 2.4. Typical procedure of the Reaction

PET flakes (5 g) and a certain amount of EG and catalyst were added in a 250 mL round-bottom three-neck flask with a magnetic stirring bar, thermometer, and reflux condenser. The system was preheated to specific temperatures (165−195 °C) before adding the PET flakes to shorten the time for the reaction temperature. After a specified time interval, the round-bottom flask was quenched quickly in iced water. Next, boiling water (200 mL) was added, so that the mixture was filtered into solid and liquid phases. The solid phase contained unreacted PET, insoluble oligomers, and catalyst. Therefore, the solid was extracted with boiling water, which was mixed with EG, catalyst, and monomer [22,23]. Undepolymerized PET was collected and dried for weighing. PET depolymerization was calculated by the following Equation (1):(1)$$\mathrm{PET}\;\mathrm{depolymerization}=\frac{W_{0}{\;-\;W}_{1}}{W_{0}}\times 100\%\;$$
where W_0_ and W_1_ represent 5 g of PET flakes and the mass of undepolymerized PET, respectively. The filtrate was stored in a refrigerator at 0 °C for 24 h. The white crystalline of BHET was obtained from the filtrate. The BHET monomer was dried in an oven at 80 °C for 24 h in vacuum oven. The yield of the BHET product was calculated according to Equation (2):(2)$$\mathrm{The}\;\mathrm{yield}\;\mathrm{of}\;\mathrm{BHET}=\frac{W_{\mathrm{BHET}}{/M}_{\mathrm{BHET}}}{W_{0}{/M}_{\mathrm{PET}}}\times 100\%\;$$
where W_BHET_, M_BHET_, and M_PET_ represent the mass of BHET, the molecular weights of BHET (254 g/mol), and the repeating unit of PET (192 g/mol), respectively.

### 2.5. Experimental Design

In order to investigate the effects between the reaction variables and optimize the BHET yield, we carried out two distinct experiments. Firstly, the conventional method, OFAT, was applied during the preliminary experiment, and four experimental variables were investigated: mass ratio of EG:PET, mass ratio of catalyst:PET, reaction temperature, and time. From the results of the OFAT, the optimization of four experimental factors (mass ratio of EG:PET, mass ratio of catalyst:PET, temperature, time) was conducted, aiming the highest yield of the BHET monomer.

The optimization of reaction conditions for PET glycolysis was carried out using RSM-onBBD technique. The coded and uncoded values of independent variables and levels in the experimental design are listed in Table 1. BBD requires an experiment number according to N = k^2^ + k + C_p_, where k is the factor number and C_p_ is the replicate number of the central point [31]. A total of 15 three-level-three-factor BBD and responses were applied based on Design Expert software (version 13, Statease Inc., Minneapolis, MN, USA). To eliminate system errors, the application was completely randomized and the center point was replicated three times to check the reproducibility [32].

### 2.6. Characterization

X-ray diffraction (XRD) patterns for oyster shell waste according to calcination treatment were analyzed using Malvern Panalytical Empyrean with Cu-Kα radiation (λ = 0.1541 nm). The operating voltage and current were 40 kV and 40 mA, respectively. Analysis was conducted over a 2θ range of 20 to 80°, with a step size of 0.028°. The morphology and elemental analysis of calcined catalysts were analyzed by field-emission scanning electron microscopy (FE-SEM) which was performed on the TESCAN, MIRA3, after coating with Pd/Pt to increase the conductivity of catalysts. The micrographs were performed at electron acceleration voltage of 5.0 kV. Elemental analysis by energy dispersive spectroscopy (EDS) was conducted at electron acceleration voltage of 15 kV. XRD patterns of the main product were analyzed at the same conditions in a range 2θ from 10° to 60°. Fourier transform-infrared spectroscopy (FT-IR) analysis was performed on Bruker alpha 2 (Bruker, Germany) using the attenuated total reflection technique (ATR), in the range of 4000–500 cm^−1^. The ^1^H- and ^13^C-NMR spectra of the main products were recorded on a Bruker500 (Bruker, Germany) spectrometer operating at 500 and 125 MHz, respectively, and all proton and carbon chemical shifts were measured relative to internal residual Dimethyl sulfoxide-d6 (99.5% DMSO-D6) from the lock solvent. Gas chromatography-mass spectrometry (GC-MS) was performed using Agilent 7890B-5977B GC/MSD. Thermal decomposition of the BHET and PET was evaluated by thermogravimetric analysis (TGA) on SDT650 (TA instrument, New castle, DE, USA) under nitrogen atmosphere at 10 °C/min from 25 °C up to 600 °C. Differential scanning calorimetry (DSC) measurement of the BHET and PET was performed using a DSC 2500 (TA instrument, USA) by heating samples from −90 °C to 200 °C at a rate of 10 °C/min under N_2_.

## 3. Results and Discussion

### 3.1. Preparation of PET Glycolysis Catalyst

PET depolymerization through glycolysis cleavage of the ester bond can be propelled by using catalysts including metal-based ceramics, organocatalysts, enzymes, and biomass-derived materials [33]. Typically, metal salts have been investigated as eco-friendly catalysts for PET glycolysis reactions due to their reusability. High temperatures and extended reaction times in the presence of a transesterification catalyst, such as a metal acetate, are required to achieve reasonable conversion efficiency. While zinc acetate is considered the benchmark for the formation of BHET, other metal salts (chloride, carbonate, sulphate) have also been reported to produce BHET via PET glycolysis [33]. Recently, a biomass-derived catalyst was demonstrated as a glycolysis catalyst, achieving 79% yield of the recrystallized product [34]. We adopted oyster waste as a biomass-derived catalyst by removing organic foreign materials and converting CaCO_3_ to CaO. Figure 2 shows how PET waste can be chemically recycled into resource material (BHET) by using a biomass-derived catalyst for closed-loop utilization.

The XRD patterns of the oyster shells before and after calcination are shown in Figure 1a. After calcination, CaO transformed from CaCO_3_ (oyster shells) was identified. The diffraction peaks correspond to cubic CaO, which are consistent with the International Center for Diffraction Data (ICDD) no. 00-043-1001. The narrow and intense peaks observed in the patterns show the high crystallinity and purity of the CaO catalyst without any other phase. Figure 1b shows the morphology and size of the particles of calcined oyster shells. These particles had an average edge length of 20 μm with a regular polynomial shape, unlike the uncalcinated oyster shell in Appendix A. From the voids on the surface, it is postulated that the calcined particles developed a greater surface area than the oyster shell by releasing CO_2_. At the higher magnification in Figure 1b, the surfaces of the particles also showed traces of cracking on all shells (Figure 1c). The EDS mapping results of the magnified region confirmed that the surface of the particles was composed of CaO (Appendix A). From the EDS results in Figure 1c and Appendix A, the calcined catalysts mainly contained approximately 70 wt% of Ca and 26 wt% of O with trace amounts of C, Na, Mg, Al, and Si.

### 3.2. Glycolysis of PET Using Oyster-Waste-Modifed Catalyst

As a biomass-derived catalyst in PET glycolysis, we used CaO powder converted from oyster shell waste, which was collected from oyster farms in coastal areas. Figure 2a shows the final product, BHET, after transesterification in ethylene glycol at 195 °C. XRD, FT-IR, and NMR were also conducted to investigate the structure and purity of the main product obtained from PET glycolysis using oyster shell as a catalyst. From the XRD patterns of the BHET product in Figure 2b, the diffraction peaks at 2θ = 6.89, 13.78, 16.43, 23.36, 27.35 were matched by α-BHET (ref. code: 00-053-1689) [35]. The FT-IR spectra of the BHET and PET are shown in Figure 2c.

The BHET spectrum presented a characteristic absorption peak indicating that the components have bands such as 2955.78 cm^−1^ and 2872.71 cm^−1^ (vibration of alkyl C-H group), 1709.54 cm^−1^ (C=O stretching vibration), and 1406.60 cm^−1^ (para-substitute group on a benzene ring), respectively. The strong bands around 3280.49 cm^−1^ are characteristic of the hydroxyl stretching of BHET, which did not appear in PET. This spectrum is consistent with previously reported BHET [29,36]. As shown in Figure 2d, proton (^8^H) and carbon (^13^C) NMR spectroscopies were performed to evaluate the chemical structure of BHET. The structure of BHET was symmetrical with respect to the benzene group, and protons of functional groups present on both sides had frequencies in the same region. The signal at δ = 8.13 ppm (peak 1) shows the presence of four aromatic proton peaks of the benzene ring. The signals at δ = 4.33 ppm (peak 2) and δ = 3.72 ppm (peak 3) indicate the protons of the methylene group near the ester group and the hydroxyl group, respectively. The signal at δ = 4.96 ppm (peak 4) was characteristic of the presence of the hydroxyl group (-OH). The peaks of DMSO and H_2_O represent solvent and water contaminant, respectively. The carbon spectrum of the BHET is shown in the lower graph of Figure 2d. The signal at δ = 165.65 ppm (peak 3) indicates the carbon of the carbonyl group. The signals at δ = 134.24 ppm (peak 2) and δ = 129.99 ppm (peak 1) indicate the carbon of the aromatic group. The signals at δ = 165.65 ppm (peak 3), δ = 67.50 ppm (peak 4) and δ = 59.46 ppm (peak 5) indicate the carbon of the carbonyl and methylene groups, respectively. Therefore, these results, confirmed by the BHET monomer with high chemical purity, are in agreement with those of previous studies [29]. In addition, the mass spectrum of the BHET obtained by GC/MS analysis was identical to its standard (Appendix A). The intense peaks at *m*/*z* 211.1, 193.1, 167.1, 149.1, 121.1, 104.1, and 76.1 are related to the monomer BHET (M.W. 254 g/mol). The characteristic fragments in the mass spectrum for BHET are given in Appendix A.

The TGA curves showed distinctive thermal decomposition behavior between BHET and PET (Appendix A). While PET showed a single weight loss at 433 °C, two weight loss regions appeared at 285 °C and 435 °C in the case of BHET. The first weight loss was due to the thermal decomposition of the BHET product. The second weight loss can be ascribed to the re-polymerization of BHET to PET, which corresponds to the elimination of hydroxyl formic acid ester [37,38]. The DSC curves also confirmed the BHET as the final product (Appendix A). The DSC curve of the PET material showed a typical glass transition temperature (T_g_) at 77.25 °C. On the other hand, one sharp endothermic peak was observed at 111.32 °C, which is in accordance with the known melting point of BHET [15]. From these results, the BHET product was verified as featuring highly purified monomer without additional endothermic or exothermic peaks.

### 3.3. Optimal Parameters on PET Glycolysis

In order to investigate the main independent process parameters and their ranges for the design of the experiments, the preliminary experiments were carried out through OFAT (one-factor-at-a-time). The effect of the reaction conditions on the PET depolymerization and the yield of BHET was studied and the results are shown in Figure 3. As shown in Figure 3a, the PET depolymerization and the BHET yield were significantly increased when the mass ratio of EG to PET was increased from 1 to 5 in the mass ratio of the catalyst to PET of 1 wt% at 195 °C for 1 h. According to previous studies [23,39], an excessive amount of EG is required for glycolysis reactions. When the mass ratio of EG to PET was 5, the yield of BHET reached 68.4%, which showed an optimal value. Thus, we applied this ratio of EG to PET in all the experiments. The slight decrease in the yield after the optimal result is considered to be due to the decreased ratio of catalyst to EG. The effect of the catalyst amount on PET depolymerization and the BHET yield is shown Figure 3b. Both PET depolymerization and BHET yield were increased up to 1 wt% of the catalyst. When the mass ratio of the catalyst to PET was increased from 0.2 to 1 wt%, PET depolymerization increased from 74.0 to 91.4 wt%, the BHET yield increased from 58.1 to 68.6%. When the catalyst content for PET was 1 wt%, the BHET yield reached its highest level, and then gradually decreased.

From the results in Figure 3c,d, it can be observed that PET depolymerization and the BHET yield rose obviously with the increase in time and temperature. The PET depolymerization was about 91.4% and the BHET yield reached 68.6% at 195 °C for 1 h. These results are comparable to Na_2_CO_3_ catalyst [23].

### 3.4. Determination of the Regression Model

The regression model used in RSM is usually a quadratic polynomial equation to predict the response as a function of independent variables. In this study, three independent variables (reaction temperature, reaction time, catalyst:PET mass ratio) were investigated, as shown in Table 1. The equation of a quadratic polynomial can be expressed as:(3)$$Y={\;\beta }_{0}+\sum_{i}\beta_{i}x_{i}+\sum_{i}\beta_{\mathrm{ii}}x_{i}^{2}+\sum_{i<j}\sum_{j}\beta_{\mathrm{ij}}x_{i}x_{j}+\varepsilon$$
where Y is the predicted response, β_0_ is a constant, and β_i_, β_ii_, and β_ij_ are linear, quadratic, and interactive coefficients, respectively. X_i_ is the independent variables in these cases (X_1_ = reaction temperature, X_2_ = time, X_3_ = mass ratio of catalyst:PET) [40]. The polynomial coefficients were estimated by least square fitting method [41,42]. Based on the single-factor analysis, BBD with three factors in three levels and experimental yield was presented in Table 2. After evaluating the fit statistics of the regression models among linear, 2F1, and quadratic in Table 3, the quadratic model showed the highest R^2^ value (0.989), adjusted R^2^ (0.970), and predicted R^2^ (0.830). Furthermore, the predicted R^2^ was in reasonable agreement with the adjusted R^2^, since the difference was less than 0.2. A calculated quadratic model expressing the relationship between the f BHET yield and the reaction conditions of the three independent factors i shown by Equation (4):
(4)$$Y=15.49+17.36X_{1}+13.96X_{2}+9.11X_{3}+7.64X_{1}X_{2}+5.54X_{1}X_{3}+3.11X_{2}X_{3}+7.81{X_{1}}^{2}+1.50{X_{2}}^{2}-0.025{X_{3}}^{2}$$

Adequate precision measures the signal-to-noise ratio; a value greater than 4 is desirable. A ratio of 24.13 for a quadratic model indicates an adequate signal. Consequently, this regression model can be used to predict the response (BHET yield) in the design space [26]. A validation of the model was performed using analysis of variance (ANOVA), and the results for the coded levels are shown in Table 4. Model terms were deemed significant if the p-values were less than 0.05. Furthermore, large F-values also suggested a significant effect. The ANOVA of this model stated that the model was highly significant, as the *p* value for the model was <0.0002. The F value of the model was 50.59, implying that the regression model was significant. There was only a 0.02% chance that the F value could occur due to noise.

Residual analysis, which represents the difference between the experimental and predicted values by regression equation, was displayed in order to verify the suitability degree of the overall model (Appendix A). From the normal probability plot of the standardized residuals and the scatter plot of the given standardized residuals and predicted values, it can be assumed that the selected model is appropriate for glycolysis reaction with the homoscedasticity of error because these residuals were randomly spread within a horizontal band within ±3. The results of the experimental values were in good agreement with the data predicted by the model in Equation (4) (Appendix A).

Figure 4 shows the main effects of all the independent variables. Based on the response of the main factor’s effect, the reaction temperature showed the greatest influence on the BHET yield, followed by the reaction time and the mass ratio of catalyst:PET, respectively. According to the coefficient value of the main effect from the regression model, the coefficient of X_1_(17.36) was higher than those of X_2_(13.96) and X_3_(9.11) in Equation (4). By evaluating each coefficient determined by the p-value in the ANOVA, the terms of the main effect (A, B), the interaction effect (AB, AC), and the quadratic effect (A^2^) were significant in their ranges for the yield of BHET. Eliminating the non-significant terms (BC, B^2^, C^2^), the modified model can be shown through Equation (5):(5)$$Y=16.20+17.36X_{1}+13.96X_{2}+9.11X_{3}+7.64X_{1}X_{2}+5.54X_{1}X_{3}+7.72{X_{1}}^{2}$$

By applying the modified equation, higher significant values were obtained with *p*-values <0.0001 and an adequate precision of 28.13. Accordingly, the model expressed by Equation (5) can accurately predict the yield of monomer under any experimental conditions.

### 3.5. Response Surface Analysis

Three-dimensional response surface plots present graphical descriptions of the regression equation as a smooth surface for the optimization of reaction conditions. In this study, one variable was fixed at the center level, and the other two variables were varied within the experimental range in order to analyze the main and interaction effects between the factors. The contour plots and the corresponding 3D response surface plots between the two variables and the BHET yield are shown in Figure 5. The interaction effect between reaction temperature and reaction time at a constant mass ratio of catalyst to PET on the BHET yield is shown in Figure 5a,b. Generally, the BHET yield had a tendency to increase with the other factors while either the reaction temperature or the time was increased. The same trend can be found in Figure 5b. These results correspond to those of the single-factor analysis. For a constant reaction time, the BHET yield was slightly increased with the mass ratio of catalyst:PET (*w*/*w*) at any fixed temperature (Figure 5c,d). This was in agreement with previous OFAT data. These results could be comparable to the best conditions for ostrich eggshells calcined with 1 wt% catalyst at 192 °C for 2 h [29]. The interaction between reaction time and mass ratio of catalyst:PET is shown in Figure 5e,f. These results indicated that the interaction effect was lower compared to other combinations. From the ANOVA of the RSM, the X_2_X_3_ term showed a less significant effect on RSM in comparison to X_1_X_2_ and X_2_X_3_.

### 3.6. Optimization of PET Glycolysis by RSM

According to the experimental results and response surface analysis the optimization of PET glycolysis was conducted through the design expert software. To discover the optimal conditions for the maximum response variable and consider the economic and environmental aspects, the reaction temperature and time were selected as ’in the range’ and the mass ratio of catalyst to PET was determined as ‘maximize’. An optimal single value for the operating conditions suggested by the software was obtained: reaction temperature = 195, time = 44.7, catalyst = 1, respectively. To verify the model appropriately, three experimental runs for approximately predicted optimum conditions (reaction temperature = 195, time = 45, catalyst = 1) were executed. Accordingly, the average BHET yield was 64.98%, which was superior to the value reported in a previous study, in which CaO was used as the glycolysis catalyst [29]. As a result of the comparison with the RSM, the value of the difference was 0.1% from the predicted yield of 64.88%. Consequentially, the modified regression model of Equation (5) was suitable for the estimation of BHET yield in the investigated ranges.

## 4. Conclusions

In conclusion, PET glycolysis with an oyster-shell-derived catalyst was experimentally and statistically investigated by focusing on the three factors of interest (mass ratio of catalyst to PET, temperature, and time). From the OFAT experiments, the BHET yield increased when the mass ratio of EG to PET and catalyst to PET and the reaction time were increased. The equilibrium yield of the BHET was about 68.6% with the reaction conditions: mass ratio (EG to PET) of 5 and 0.01 (catalyst to PET) at 195 °C for 1 h. The relationship between the BHET yield and the reaction parameters (temperature, time, mass ratio of catalyst to PET) were mathematically investigated according to the BBD matrix and a quadratic regression equation. As a result of optimizing the PET glycolysis from the contour and response surface plots, the single-factor analysis demonstrated that the main variables showed an ascending order of importance: reaction temperature > time > mass ratio of catalyst to PET. Considering the economic and environmental aspects, the optimal conditions were determined as follows: 195 °C, 45 min., 1 wt% for reaction temperature, time, and mass ratio of catalyst to PET, respectively. The experimental average BHET yield under optimum conditions was 64.98%, which was in good agreement with the predicted value of 64.88%. These findings pave the way for an alternative method to promote chemical recycling of PET waste by using oyster-shell waste with optimal processing conditions for energy- and material-efficient plastic recycling.

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
