# Peer review of "Optimizing PET Glycolysis with an Oyster Shell-Derived Catalyst Using Response Surface Methodology"

_polymers, 2022, doi:10.3390/polym14040656_

Round 1

Reviewer 1 Report

Kim and co-workers present an interesting manuscript on the use of CaO derived from oyster shells as catalyst for the glycolysis of PET using an experimental design approach.

  • Graphical abstract: please check the spelling of BHET;
  • Introduction: here the possibility of depolymerizing PET using the cutinase biocatalyst should be mentioned as it is one of the most uprising approaches currently investigated;
  • Some figure's quality should be improved as they appear a bit blurry in the current version of the manuscript;
  • An English spelling check is required throughout the manuscript.

I recommend publication of this manuscript after the consideration of the above mentioned minor comments.

Author Response

“Kim and co-workers present an interesting manuscript on the use of CaO derived from oyster shells as catalyst for the glycolysis of PET using an experimental design approach.

-> We thank the reviewer for his/her valuable comment. We made all the correction point-by-point as the reviewer suggested.

  • Graphical abstract: please check the spelling of BHET

-> We corrected the spelling in the graphical abstract.

  • Introduction: here the possibility of depolymerizing PET using the cutinase biocatalyst should be mentioned as it is one of the most uprising approaches currently investigated

-> We added the suggested reference for the enzyme-based glycolysis in page 5 and highlighted in blue.

  • Some figure's quality should be improved as they appear a bit blurry in the current version of the manuscript

-> We modified the following images: Scheme 1, Fig. 1, Fig. 2, Fig. 3.

  • An English spelling check is required throughout the manuscript.

-> We went through spelling check and made corrections.

Reviewer 2 Report

Comments to Auteurs

This manuscript demonstrates the PET depolymerization into bis(2-hydroxyethyl) terephthalate (BHET) via glycolysis using an eco-friendly and economically competitive catalyst, derived from oyster shell waste. The inter-related effects of reaction parameters (time, temperature, catalyst:PET ratio) on the yield of BHET were also determined for optimizing the depolymerization process.

The manuscript is presented well and performed all the necessary experiments or characterization for the clarity.  The study provides an effective strategy for the PET depolymerization into BHET via glycolysis. However, there are some typos and points which must be clear before publication.

For example,

  • The resolution of TOC figure can be increased, specially the chemical structures.
  • Please explain in couple of sentences that why your work is distinguished with that performed by the Yunita at al.
  • In line 123, please add some description why the shells were heated at 1000 C.
  • In line 127, replace “The depolymerization of PET is a glycolysis process” with “Glycolysis is the depolymerization process”.
  • In line 128 “oligomer was” should be “oligomers were”.
  • In Section 2.4, the subtitle “Glycolysis Reaction” can be “Typical procedure of the reaction”.
  • Scheme 1, should be like this way

  • In the caption of scheme 1, please remove “recycling circle”. because the scheme is not related to recycling or circle.
  • Please check figure 1c for the area of Ca.
  • Please add the SEM image (if you already have) of uncalcinated powder for better compression in figure 2b.
  • Remove the sentence (line 298-299), because it is repeated several time “CaO catalyst from the oyster shell was used in the depolymerization of PET in the mass ratio of EG to PET of 5 299 at 195 °C for 1 h”.
  • In line 302, rephrase the sentence “1 wt%, it was observed that the conversion of PET from 74.0 to 91.4 wt%”.
  • Similarly, rephrase the sentence (line 306-308) “. At 195 °C and after only 1h, the conversion of PET into BHET is about 91.4% and the yield of monomer reached equilibrium state (68.6%), which is comparable to Na2CO3 catalyst”
  • In figure 3a-d, conversion of PET (on y-axis) should be “%depolymerzation”
  • I would suggest that replace the” conversion of PET” in the manuscript to PET depolymerization. It is because conversion of PET mean conversion to many things. Here you mean conversion of PET is the depolymerization.
  • In figure 4, label all the plots with “a” “b” and “c”.
  • In figure 5b,d,f “the yield of BHET (%)” should be “BHET Yield (%)”.
  • In conclusion, line 419, remove the word “reaction” before temperature in parenthesis.

Author Response

“The manuscript is presented well and performed all the necessary experiments or characterization for the clarity. The study provides an effective strategy for the PET depolymerization into BHET via glycolysis. However, there are some typos and points which must be clear before publication.”

-> We thank the reviewer for raising good points. We revised the manuscript point-by-point as the reviewer suggested.

i) The resolution of TOC figure can be increased, specially the chemical structures.

-> We corrected the reaction mechanism and enhanced the resolution of the image.

ii) Please explain in couple of sentences that why your work is distinguished with that performed by the Yunita at al.

-> We added the following sentence to clarify the difference in page 6 and highlighted in blue.

“However, statistical studies of the optimization of reaction conditions on the type of by-products were insufficient. Besides, it is challenging to collect and process by-products of ostrich, chicken egg shells considering further scale-up of the resources in industrial usage.”

iii) In line 123, please add some description why the shells were heated at 1000 C.

-> The calcium carbonate (oyster shell) can be transformed to calcium oxide by removing carbon dioxide at high temperature. We revised the sentence in page 7 and highlighted in blue.

 “In order to transform the phase, the shells were heated at 1,000 °C in the air with a ramping rate of 20 °C min-1 for 5 h.”

iv) In line 127, replace “The depolymerization of PET is a glycolysis process” with “Glycolysis is the depolymerization process”.

-> We replaced the sentence as the reviewer suggested in page 7 and highlighted in blue.

v) In line 128 “oligomer was” should be “oligomers were”.

-> We corrected it and highlighted in blue.

vi) In Section 2.4, the subtitle “Glycolysis Reaction” can be “Typical procedure of the reaction”.

-> We revised it according to the reviewer’s comment in page 7 and highlighted in blue.

vii) Scheme 1, should be like this way

-> We modified the reaction mechanism in scheme 1.

viii) In the caption of scheme 1, please remove “recycling circle”. because the scheme is not related to recycling or circle.

-> We removed it.

ix) Please check figure 1c for the area of Ca.

-> We modified the area in Fig. 1B to fit the enlarged image in Fig. 1C.

x) Please add the SEM image (if you already have) of uncalcinated powder for better compression in figure 2b.

-> We added the uncalcinated powder in supporting information (Fig. S1) and added the phrase in page 12 as follows and highlighted in blue.

 “These particles have the average edge length of 20m with a regular polynomial shape with a regular polynomial shape unlike uncalcinated oyster shell in Fig. S1.

xi) Remove the sentence (line 298-299), because it is repeated several time “CaO catalyst from the oyster shell was used in the depolymerization of PET in the mass ratio of EG to PET of 5 299 at 195 °C for 1 h”.

-> We removed the sentence.

xii) In line 302, rephrase the sentence “1 wt%, it was observed that the conversion of PET from 74.0 to 91.4 wt%”.

-> We revised the phrase and highlighted in blue.

In page 15, the modified sentence is as follows: “…, PET depolymerization increased from 74.0 to 91.4 wt%, the yield of BHET increased from 58.1 to 68.6 %.”

xiii) Similarly, rephrase the sentence (line 306-308) “. At 195 °C and after only 1h, the conversion of PET into BHET is about 91.4% and the yield of monomer reached equilibrium state (68.6%), which is comparable to Na2CO3 catalyst”

-> We revised the manuscript as follows and highlighted in blue.

In page 16, the modified sentence is as follows: “PET depolymerization is about 91.4% and the yield of BHET reached 68.6% at 195 °C for 1h, this result is comparable to Na2CO3 catalyst.”

xiv) In figure 3a-d, conversion of PET (on y-axis) should be “%depolymerzation”

-> We modified “conversion of PET(%)” to “depolymerization (%)”.

xv) I would suggest that replace the” conversion of PET” in the manuscript to PET depolymerization. It is because conversion of PET mean conversion to many things. Here you mean conversion of PET is the depolymerization.

-> We changed the word “conversion of PET” to “PET depolymerization”.

xvi) In figure 4, label all the plots with “a” “b” and “c”.

-> We labelled them.

xvii) In figure 5b,d,f “the yield of BHET (%)” should be “BHET Yield (%)”.

-> We modified it.

xviii) In conclusion, line 419, remove the word “reaction” before temperature in parenthesis

-> We removed it.
